# OpenReview forum: "Your Task May Vary: A Systematic Understanding of Alignment and Safety Degradation when Fine-tuning LLMs"
_ICLR.cc/2025/Conference — Submitted to ICLR 2025_

### Official Review · Reviewer_AWsZ · 2024-11-02

**Soundness:** 2
**Presentation:** 3
**Contribution:** 2
**Rating:** 5
**Confidence:** 3

**Summary:**

The paper studies how upstream datasets affect the durability of safety guardrails during downstream fine-tuning in large language models.  The authors conjecture that safety guardrails are more durable when the upstream dataset is more diverse and less similar than the downstream dataset, which is verified using the LLAMA2-7B-CHAT model under various datasets.

**Strengths:**

The paper shows that a malicious actor can exploit the similarity of upstream and downstream datasets to weaken the safety guardrails. Hence, it is crucial to protect the privacy and improve the diversity of the upstream dataset. This is a meaningful observation with important real-world implications for improving LLM safety.

**Weaknesses:**

The observation regarding how improving the privacy and diversity of the upstream dataset can help with safety is somewhat expected and has already been recognized in the literature to a great extent. The metrics of similarity and diversity used in the paper are both adapted from recent work. For example, it was observed in He et al. (2024) that selecting benign examples that are most similar to known harmful data can improve the attack success rate significantly. The idea of protecting the privacy of training data to hinder adversarial manipulation is also well known in security and adversarial machine learning communities.

The evaluation is far from being comprehensive. In addition to the several limitations already mentioned in the paper, including using a single LLM architecture of fixed size, one important weakness is that when evaluating the impact of similarity between upstream and downstream tasks, only the cluster of list-format data is considered. Further, the paper does not consider any defense against downstream fine-tuning attacks. It is unclear if similar results or new insights can be obtained when considering other clusters of similar data or downstream tasks with some defense applied.

**Questions:**

In Section 4.2, the paper shows that low-similarity data is more diverse than high-similarity data. Why is this always the case? The diversity only concerns the upstream data, while the similarity measure involves both the upstream and downstream datasets.

---

> ### Author Response · Authors · 2024-11-23
> **Response to Reviewer AWsZ (1 of 2)**
>
> Thank you for your time and valuable feedback. We appreciate your recognition of the significance of our proposed framework and its potential real-world impact on enhancing LLM safety. To your questions, we address our response in the following.
>
> **1. (Novelty)** We respectfully disagree that our findings are not novel. Our research is orthogonal to that of He et al. [1], who focus on identifying harmful data within benign datasets. In contrast, our anchor-free method (Section 3.1) aims to uncover an even more harmful subset without relying on harmful anchors. While He et al. utilize 100 harmful data points as anchors to score and rank harmfulness based on cosine similarity, our approach does not rely on harmful data at all. Instead, we concentrate on the similarity of upstream data alone. This anchor-free methodology fundamentally differs from previous studies and offers a novel perspective from similarity on the issue.
>
> Based on this, our research extends the analysis to the similarity between downstream finetuning data and upstream data, revealing a correlation between diversity and the robustness of safety mechanisms. Our findings can assist in making LLMs more trustworthy by informing the curation of datasets that enhance safety alignment and make the safety guardrail durable after finetuning.
>
> **2. (Generalizability of Model Arch/Size)**
> This is a great suggestion! We have conducted the experiments using Gemma2 (2B/9B), and the results are provided in the table below and Appendix C3. These results confirm the same phenomenon: the safety measures established by the low-similarity alignment dataset are more resilient following downstream fine-tuning. We will include full experimental results in the revision.
>
> **Table S1: The Utility/Harmfulness Before/After Downstream Fine-tuning on Gemma2-9B.**
> ||||||$~~~~~$5K||
> |-:|:-:|-:|-:|-:|-:|-:|
> |||None|Full(7.7K)|High-Sim|Random|Low-Sim|
> |**Initial**|Utility|7.95|8.05|8.02|7.83|7.90|
> ||GPT Score|3.94|1.41|1.53|1.50|1.56|
> ||GPT ASR|61.33%|6.67%|9.56%|7.67%|9.56%|
> ||||||||
> |**List**|GPT Score|4.48|3.72|3.89|3.66|**3.61**|
> ||GPT ASR|71.33%|55.33%|56.67%|59.67%|**50.33%**|
> |**Pure Bad**|GPTScore|4.61|4.59|4.59|4.58|**4.50**|
> ||GPT ASR|84.33%|86.33%|80.00%|80.33%|**79.33%**|
> ||||||||
> |**Alpaca**|Utility|5.45|5.35|5.41|5.11|4.84|
> ||GPT Score|4.65|4.60|4.67|4.68|**4.62**|
> ||GPT ASR|72.67%|72.00%|75.33%|76.00%|75.67%|
>
> **Table S2: The Utility/Harmfulness Before/After Downstream Fine-tuning on Gemma2-2B.**
> ||||||$~~~~~$5K||||$~~~~~$1K||
> |-:|:-:|-:|-:|-:|-:|-:|-|-:|-:|-:|
> |||None|Full(7.7K)|High-Sim|Random|Low-Sim||High-Sim|Random|Low-Sim|
> |**Initial**|Utility|7.09|7.11|7.50|7.43|7.21||7.33|6.98|7.32|
> ||GPT Score|4.19|2.03|2.21|1.96|2.27||2.80|2.83|2.67|
> ||GPT ASR|65.00%|18.33%|22.78%|19.00%|23.78%||37.67%|36.67%|34.89%|
> ||||||||||||
> |**List**|GPT Score|4.58|4.21|4.37|4.24|**4.21**||4.49|4.43|**4.43**|
> ||GPT ASR|70.00%|63.33%|67.00%|64.00%|**62.67%**||70.00%|67.33%|**63.00%**|
> |**Pure Bad**|GPTScore|4.45|4.48|4.55|4.48|**4.42**||4.53|4.49|**4.46**|
> ||GPT ASR|79.67%|79.33%|79.00%|78.00%|**74.67%**||82.00%|79.67%|**79.33%**|
> ||||||||||||
> |**Alpaca**|Utility|5.66|5.64|5.14|5.30|5.50||5.52|5.45|5.64|
> ||GPT Score|4.53|4.46|4.52|**4.45**|4.49||4.59|4.54|**4.46**|
> ||GPT ASR|69.00%|66.67%|67.00%|66.67%|**68.33%**||73.33%|68.00%|**68.00%**|
>
>
> *References:*
>
> [1] He et al. What is in Your Safe Data? Identifying Benign Data that Breaks Safety (COLM 2024)

---

> ### Author Response · Authors · 2024-11-23
> **Response to Reviewer AWsZ (2 of 2)**
>
> **3. (Why Only List Format is Considered)** We need to emphasize that the list format serves as the representative data to assess the impact of benign fine-tuning attacks on LLM safety. Fine-tuning attacks are categorized into harmful and benign types: harmful fine-tuning typically uses purely harmful datasets, whereas, as shown by He et al., and also can be seen in our experiments in Section 3.1, benign fine-tuning on downstream list data has the most substantial impact on safety. Therefore, the list format serves as the representative one, effectively and efficiently acting as the benign counterpart to purely harmful datasets in harmful fine-tuning attacks.
>
> Furthermore, in our new experiment (available in Appendix C2), we have fine-tuned each cluster, and our results demonstrate that this insight can also be obtained from other clusters (of different question types).
>
> **4. (Downstream Defense)** This is a great suggestion! We will provide an experiment using fine-tuning stage defense (e.g., SafeInstr [2], Backdoor Enhanced Alignment [3]) with our aligned models. If the reviewer has a specific method that would like to be discussed, please let us know; we are happy to provide the results and discussion. Due to the limited time of the discussion period, we will provide these results in the next few days.
>
> **5. (Diversity Measurement)**
> Thank you for pointing out this concern. We need to acknowledge that it is not always the case that low-similarity is more diverse than high-similarity data. In some extreme cases, the opposite scenario may occur: for example, when the upstream dataset is *highly self-similar* (e.g., consisting of a large number of repeated data), it can show low similarity to downstream tasks while also being low in diversity. However, we need to emphasize this is a very unrealistic scenario. The main contribution of our work is we find that *when the dataset used for upstream safety alignment is highly similar to the downstream fine-tuning data, safety alignment can be severely compromised by fine-tuning attacks*. Also, it is intuitive to find that upstream data that shows high similarity to downstream data typically lacks diversity, so enhancing the diversity of the upstream safety alignment data is a promising way to fortify safety guardrails against potential threats, therefore, enhancing diversity starting from high-similarity data is inherently a process of lowering similarity.
>
>
> *References:*
>
> [2] Bianchi et al. Safety-Tuned LLaMAs: Lessons From Improving the Safety of Large Language Models that Follow Instructions. (ICLR 2024)
>
> [3] Wang et al. Mitigating Fine-tuning based Jailbreak Attack with Backdoor Enhanced Safety Alignment

---

> > ### Comment · Reviewer_AWsZ · 2024-11-24
> >
> > I would like to thank the authors for the detailed rebuttal, which has addressed some of my concerns. I have updated my rating. However, I remain unconvinced that the main findings of the paper are surprising. It is intuitive that selecting benign examples less similar to downstream data can help reduce the attack success rate but may also hurt the utility due to its narrow scope, as shown in Table 2. Having some experiments to verify this intuition is interesting but does not provide concrete guidance. In particular, is it better to choose a Random subset or a Low-Sim subset? A deep exploration of this tradeoff in the context of LLM fine-tuning is lacking.

---

> > > ### Author Response · Authors · 2024-11-24
> > >
> > > We thank the reviewer for the prompt response! We are glad our response has addressed some of your concerns. While our current work focuses on demonstrating the correlation between dataset similarity and guardrail durability, we acknowledge the challenge in choosing between Random and Low-Sim alignment in practice. However, we would like to emphasize that our study aims to understand how upstream/downstream similarity affects guardrail durability. The experiment was designed with the assumption that we had access to the downstream task, enabling us to form a Low-Sim alignment dataset compared to High-Sim settings. The "*Random*" setting simulates a *practical* scenario of how a standard dataset curation process would perform. Although the diversity analysis in Figure 4 shows slight differences between Random and Low-Sim, it provides insight into how to curate a more durable safety alignment dataset, given the same alignment size.
> > >
> > > That being said, we acknowledge that the current version might confuse readers regarding the choice between Random and Low-Sim alignment datasets, and we will revise our paper to clarify this point. Once again, we thank you for your time and constructive feedback!

---

> ### Author Response · Authors · 2024-11-29
> **Results on Downstream Defense**
>
> We again appreciate the reviewer’s effort in reviewing our paper. We have now completed experiments that combine our approaches with downstream defenses. The results are shown in the table below. In this additional experiment, we consider two defense strategies: SafeInstr [2] and Backdoor Enhanced Alignment (BEA) [3]. These methods achieve defense by mixing a certain proportion of safety alignment data into the fine-tuning task, thereby allowing the established safety guardrails to persist.
>
> For SafeInstr, the authors demonstrate that fine-tuning models by adding safety samples can improve their safety. We have augmented their safe instructions into the fine-tuning datasets, with safety samples comprising 10% of the PureBad/List dataset and 3% of the Alpaca dataset. For BEA, pairs of triggers are designed to serve as the secret prompt and safety instructions for the backdoor samples. During the inference phase, if the trigger is detected and the user's instructions are harmful, their impact will be mitigated. In our experiments with BEA, we used 10% of the backdoor samples for the PureBad/List dataset and 1% for the Alpaca dataset. Our results suggest that additional protection mechanisms reinforce safety guardrails, strengthening them against fine-tuning attacks. We will include full experimental results in the revision.
>
> We hope this additional experiment addresses any outstanding concerns the reviewer might have and assists in finalizing the review rating.
>
> **Table S3: The Harmfulness Before/After Downstream Fine-tuning (with Defense) on Llama2-7B .**
> ||||$~~~~~$1K||
> |-:|:-:|-:|-:|-:|
> |||High-Sim|Random|Low-Sim|
> |**Initial**|GPT Score|2.05|2.02|2.21|
> ||GPT ASR|18.67%|20.33%|23.17%|
> ||||||
> |**List**|GPT Score|4.82|4.68|**4.35**|
> ||GPTASR|79.00%|74.33%|**71.67%**|
> |**List** + *SafeInstr*|GPT Score|3.84|3.67|**3.49**|
> ||GPT ASR|60.33%|55.67%|**53.67%**|
> |**List** + *BEA*|GPT Score|3.00|3.33|**2.98**|
> ||GPT ASR|43.33%|50.33%|**42.00%**|
> ||||||
> |**Pure Bad**|GPT Score|4.63|4.65|**4.63**|
> ||GPT ASR|76.33%|74.67%|**74.33%**|
> |**Pure Bad** + *SafeInstr*|GPT Score|3.51|3.63|**3.44**|
> ||GPT ASR|57.00%|57.00%|**54.00%**|
> |**Pure Bad** + *BEA*|GPT Score|2.59|3.08|**2.55**|
> ||GPT ASR|33.00%|46.33%|**33.00%**|
> ||||||
> |**Alpaca**|GPT Score|3.54|3.73|**3.33**|
> ||GPT ASR|53.67%|57.67%|**48.00%**|
> |**Alpaca** + *SafeInstr*|GPT Score|2.48|2.50|**2.39**|
> ||GPT ASR|30.00%|30.33%|**27.33%**|
> |**Alpaca** + *BEA*|GPT Score|2.02|1.82|**1.54**|
> ||GPT ASR|24.67%|19.00%|**10.67%**|

---

> ### Author Response · Authors · 2024-12-02
> **Look forward to Discuss with You**
>
> Dear Reviewer AWsZ,
>
> We would like to thank you again for your valuable time and efforts spent reviewing our paper and helping us improve it. As the discussion period is closing, it would be very much appreciated if you could once again help review our responses and let us know if these resolve your remaining concerns. We are eager to answer any further questions you might have. We strive to improve the paper consistently, and it is our pleasure to have your feedback!
>
> Yours Sincerely,
>
> Paper #950 Authors

---

### Official Review · Reviewer_5v66 · 2024-11-03

**Soundness:** 3
**Presentation:** 3
**Contribution:** 2
**Rating:** 5
**Confidence:** 4

**Summary:**

This paper investigates how to make LLMs' safety guardrails more durable after downstream fine-tuning. The authors find that safety guardrails are more fragile when the upstream and downstream datasets are similar but more durable when they are dissimilar and diverse.

**Strengths:**

1. This paper provides a new insight about the fragility of safety guardrails, which stems from the high similarity between upstream and downstream tasks.
2. Instead of using constraint loss functions, this paper proposes a new direction: data-based approach for fine-tuning attacks.

**Weaknesses:**

1. The real-world applicability of the findings is questionable. In the experiments, the authors have access to the complete downstream data, yet the safety improvement is modest. According to Table 2, the GPT score is about 0.1 - 0.4 lower than random selection, and the GPT ASR is around 5% lower. This gap is likely to be smaller in real-world scenarios, where downstream data from attackers are not accessible.
2. The paper concludes that low-similarity datasets are more diverse than high-similarity ones, which is intuitively understandable and not a particularly novel or interesting insight. Moreover, the paper does not provide new methods or insights on how to obtain a more diverse safety-alignment dataset.
3. The authors should include constraint loss based baselines in their experiments to better demonstrate the effectiveness of data similarity in defending against fine-tuning attacks.

**Questions:**

In table 1, the clustering algorithm is k-means. As we know k-means contains some randomness, are the results in table 1 the average of several runs?

---

> ### Author Response · Authors · 2024-11-23
> **Response to Reviewer 5v66**
>
> We appreciate the reviewer’s effort in reviewing our paper and recognizing the new insights and directions of our work. Below, we address the weaknesses and questions posed.
>
> **1. (Applicable Scenarios)**
> We understand the reviewers' concerns. However, we respectfully disagree that this is a weakness of our experiment. As a mechanistic study, we seek to explore how the similarity between upstream and downstream data affects the safety guardrail. Notably, in Table 2, our results demonstrate consistent durability (of the safety guardrail under Low-Sim alignment) across various upstream safety alignment data sizes and different downstream tasks, with a maximum difference of 9.67% in GPT ASR. This trend can also be observed in different model architectures and sizes (see more results in Appendix C3).
>
> In addition, in real-life scenarios, even if a more significant amount of aligned data (e.g., Llama) is used, jailbreak will occur after benign fine-tuning, and our experiments also verified that one of the reasons could be that certain parts of downstream data share a high similarity with upstream alignment data. This finding suggests that the durability of safety guardrails depends on the privacy and diversity of upstream safety datasets. To our knowledge, this is the first study to investigate how fine-tuning attacks damage safety by examining the interplay between upstream alignment and downstream fine-tuning.
>
>
> **2. (More Diverse Dataset)** We thank the reviewer for raising this point. We now understand that our original description might need some clarification (we will adjust it in the revision). In our study, our core contribution is discovering and understanding how the upstream/downstream dataset similarity affects the durability of the safety guardrail, which we believe is novel and has never been studied under different LLM training stages.
>
> Based on our findings, we adopted a diversity metric [1] to measure the characteristics of the upstream dataset, and further found that when it is *highly similar* to the downstream dataset, it often lacks diversity. While it is not sufficient, it is necessary to highlight the importance of enhancing the diversity of the upstream dataset during safety alignment. That being said, one could always increase the diversity by obtaining more data; as shown in Table 2, using the *Full* alignment dataset could build a safer and more durable guardrail.
>
> **3. (Loss-based Constraint)**
> We appreciate the reviewer's comment. As pointed out by the reviewer, our proposed method is orthogonal (addressed solely from a dataset perspective) to the loss-based approach. Nevertheless, we are happy to include a loss-based constraint method in the revision.
>
> **4. (Several Runs of K-means)**
> We have revised our paper and provided the means and standard deviation of three runs (with three random states). When doing k-means clustering, we could always obtain clusters of different question types (e.g., list format, coding, summary, etc.), and we provide the experimental results on the list-format cluster as it has been considered to have the most substantial impact on safety.
>
> |||| $~~~~~~~~$Alpaca|| $~~~~~~~~$Dolly   ||
> | :-: | -: | -: | -: | -: | -: | -: |
> |             | Initial | Pure Bad | Top-100 Harmful |Cluster | Top-100 Harmful | Cluster |
> | GPT Score   | 1.13       | 4.52     | 3.57            |**4.01** (±0.2)   | 3.59    | **4.10** (±0.3)    |
> | GPT ASR (%) | 1.00%   | 72.33%   | 32.67%| **55.44%** (±12.8) | 47.00%  | **59.89%** (±9.6)  |
>
> *References:*
>
> [1] Wang et al. Diversity measurement and subset selection for instruction tuning datasets

---

> > ### Comment · Reviewer_5v66 · 2024-11-25
> >
> > Thanks to the authors for their rebuttal, which addresses some of my concerns. However, I still believe that the main findings of the paper are not particularly surprising. A diverse dataset is typically a goal for downstream tasks, and attackers can also enhance their diversity.

---

> > > ### Author Response · Authors · 2024-11-25
> > >
> > > We thank the reviewer for their prompt response and engagement with our work! We totally agree that achieving a diverse dataset is a common goal for downstream tasks, as it is crucial for enhancing generalization and improving performance on target tasks. Our intention was never to suggest that increasing diversity is risky to safety guardrails. Instead, we posit that enhancing dataset diversity is beneficial and can lead to greater robustness of safety guardrails.
> > >
> > > From the perspective of similarity, increasing diversity inherently reduces self-similarity, which helps avoid overfitting. In turn, this could diminish the likeness between upstream and downstream datasets, thereby fortifying the safety guardrails. Therefore, our findings emphasize the importance of diversifying the upstream dataset to enhance the resilience of safety alignments throughout the fine-tuning process. By reducing self-similarity and decreasing the overlap between upstream and downstream datasets, we can mitigate overfitting and improve the durability of safety mechanisms.
> > >
> > > We appreciate the reviewer's insights and hope our response clarifies our stance. We remain open to further feedback and are keen to continue refining our contributions in light of valuable discussions like this.

---

> ### Author Response · Authors · 2024-12-02
> **Look forward to Discuss with You**
>
> Dear Reviewer 5v66,
>
> We would like to thank you again for your valuable time and efforts spent reviewing our paper and helping us improve it. As the discussion period is closing, it would be very much appreciated if you could once again help review our responses and let us know if these resolve your remaining concerns. We are keen to answer any questions you might have. We strive to improve the paper consistently, and it is our pleasure to have your feedback!
>
> Yours Sincerely,
>
> Paper #950 Authors

---

### Official Review · Reviewer_ykti · 2024-11-04

**Soundness:** 2
**Presentation:** 3
**Contribution:** 3
**Rating:** 6
**Confidence:** 4

**Summary:**

The authors study the relationships between data diversity across different training stages of LLMs and its impact on the ability to weaken safety guardrails. The authors perform experiments on llama 3 architecture and conclude that 1) keeping the training data private may reduce the ability of an attacker to jailbreak the models, 2) higher diversity may also reduce the attacker's ability to jailbreak the models.

**Strengths:**

- Originality: the paper studies the impact of dataset similarity across multiple stages of training as a possible aspect that could impact the effectiveness of safety guardrails in fine-tuning. I don't know other papers focusing on this aspect.
- Clarity: the paper is clear and well written, with nice diagrams that illustrate the concepts easily.
- Quality: several empirical aspects limit the scope and validity of the analysis
- Significance: the paper is extremely relevant for the community.

- The proposed procedure is interesting and could potentially be very useful if the authors demonstrated the generalisability of its assumptions (see weaknesses)

**Weaknesses:**

- The conclusion that better safety can be achieved by obscurity about the training data, although very practical, is not a typical recommendation the security community would make.
- The generalizability of the paper's conclusions is unclear, since the work focuses on very restricted model architectures. The claims need further empirical evidence, it could be good if the authors could include in the rebuttal. Both to study the phenomenon for different architectures and model sizes
- It is unclear if the conclusions drawn depend on the choice of the jailbreaks. Indeed, an extensive suite of multiple jailbreak benchmarks should be used in order to draw solid conclusions. Furthermore, one would also need to present some adaptive forms of jailbreak generation that do account for the differences in training between models. While even this would not guarantee the absolute generalizability of the conclusions (which would remain a core limitation), it would make a much stronger empirical point.
- the statements also only hold for the considered guardrails, and it's unclear how well they generalise to other forms of guardrails.

I am happy to increase my score if the points above are well addressed with experiments and evidence.

**Questions:**

- Could the authors consider using data contamination detection procedures in order to reduce the conjecturality about section 3.1?

---

> ### Author Response · Authors · 2024-11-23
> **Response to Reviewer ykti (1 of 2)**
>
> We appreciate the reviewer’s effort in reviewing our paper and recognizing our work as unique and well-written, especially for studying the safety guardrail across different stages that have never been explored before and for pointing out its potential to be very useful. We are thrilled you provided encouraging reviews and constructive comments. Below we address the weaknesses and questions posed.
>
> **1. (Security Community View)** We thank the reviewer for raising this viewpoint. While we agree that maintaining *closed-source* data is not a sustainable long-term strategy for building robust safety guardrails, our work introduces a novel finding:  *when the data used for upstream safety alignment closely resembles the data used for downstream fine-tuning, it significantly compromises safety alignment due to fine-tuning attacks*. This sheds light on why even seemingly benign downstream fine-tuning can substantially degrade safety. Our focus is not merely on data confidentiality but on understanding the diversity and robustness of the alignment process itself, regardless of whether upstream data are obscured or disclosed.
>
> Moreover, as pointed out by the reviewer in the question, researchers studying dataset contamination have proposed promising methods for detecting whether specific data have been used during pre-training or alignment [1, 2]. In our opinion, we would consider this a potential threat, as an attacker could exploit this knowledge to collect data used for safety alignment and craft a benign-looking dataset aimed at compromising a model's safety measures. Thus, maintaining the confidentiality of the alignment dataset is merely a preliminary step in mitigating this risk. Further studies are required to address challenges like preventing membership inference attacks (MIA) on the alignment dataset. In our revision, we will include a discussion on the role of MIA defense mechanisms, thereby enhancing the comprehensiveness of our argument.
>
> **2. (Generalizability of Model Arch/Size)**
> This is a great suggestion! We have run the experiments on Gemma2 (2B/9B), and the results are provided in the table below and Appendix C3. The results show the same phenomenon: the safety guardrails created by the low-similarity alignment dataset remain more durable after downstream fine-tuning. We will include full experimental results in the revision.
>
>
> **Table S1: The Utility/Harmfulness Before/After Downstream Fine-tuning on Gemma2-9B.**
> | | | | | |$~~~~~$5K | |
> | -: | :-: | -: | -: | -: | -: | -: |
> | | | None |  Full (7.7K) | High-Sim | Random | Low-Sim |
> | **Initial**  | Utility   | 7.95 | 8.05 | 8.02     | 7.83 | 7.90 |
> | | GPT Score | 3.94 | 1.41 | 1.53 | 1.50 | 1.56 |
> | | GPT ASR   | 61.33% | 6.67% | 9.56% | 7.67% | 9.56% |
> | | |  | |  | | |
> |**List**| GPT Score | 4.48 | 3.72 | 3.89 | 3.66 | **3.61** |
> | | GPT ASR   | 71.33% | 55.33% | 56.67% | 59.67% | **50.33%** |
> |**Pure Bad**| GPT Score | 4.61 | 4.59 | 4.59 | 4.58 | **4.50** |
> | | GPT ASR   | 84.33% | 86.33% | 80.00% | 80.33% | **79.33%** |
> | | | | | |  |  |
> |**Alpaca**| Utility   | 5.45 | 5.35 | 5.41 | 5.11 | 4.84 |
> | | GPT Score | 4.65 | 4.60 | 4.67 | 4.68 | **4.62** |
> | | GPT ASR   | 72.67% | 72.00% | 75.33%   | 76.00% | 75.67% |
>
>
>
> **Table S2: The Utility/Harmfulness Before/After Downstream Fine-tuning on Gemma2-2B.**
> ||||||$~~~~~$5K||||$~~~~~$1K||
> | -: | :-: | -: | -: | -: | -: | -: | - | -: | -: | -: |
> |||None|Full (7.7K)|High-Sim|Random|Low-Sim| |High-Sim|Random|Low-Sim|
> |**Initial**|Utility|7.09|7.11|7.50|7.43|7.21| |7.33|6.98|7.32|
> ||GPT Score|4.19|2.03|2.21|1.96|2.27| |2.80|2.83|2.67|
> ||GPT ASR|65.00%|18.33%|22.78%|19.00%|23.78%| |37.67%|36.67%|34.89%|
> ||||||||||| |
> |**List**|GPT Score|4.58|4.21|4.37|4.24|**4.21**| |4.49|4.43|**4.43**|
> ||GPT ASR|70.00%|63.33%|67.00%|64.00%|**62.67%**| |70.00%|67.33%|**63.00%**|
> |**Pure Bad**|GPTScore|4.45|4.48|4.55|4.48|**4.42**| |4.53|4.49|**4.46**|
> ||GPT ASR|79.67%|79.33%|79.00%|78.00%|**74.67%**| |82.00%|79.67%|**79.33%**|
> ||||||||||| |
> |**Alpaca**|Utility|5.66|5.64|5.14|5.30|5.50| |5.52|5.45|5.64|
> ||GPT Score|4.53|4.46|4.52|**4.45**|4.49| |4.59|4.54|**4.46**|
> ||GPT ASR|69.00%|66.67%|67.00%|66.67%|**68.33%**| |73.33%|68.00%|**68.00%**|
>
>
> *References:*
>
> [1] Shi et al. Detecting Pretraining Data from Large Language Models (ICLR 2024)
>
> [2] Zhang et al. Pretraining Data Detection for Large Language Models: A Divergence-based Calibration Method (EMNLP 2024)

---

> ### Author Response · Authors · 2024-11-23
> **Response to Reviewer ykti (2 of 2)**
>
> **3. (More Benchmarks)**  We thank the reviewer for pointing out this concern. To clarify, in our experiments, we follow the same setting as Qi et al. [3], where we evaluated model safety without employing any jailbreak approach or special prompting methods to elicit harmful responses. Likewise, the jailbreak benchmark data is neither involved during alignment or downstream fine-tuning, nor is it used as an anchor point for data selection.
>
> To further substantiate our findings, we have also obtained safety results using AdvBench [4] in the following table (also available in Appendix C4), and the results are consistent with our claims.
>
> **Table S3: The Utility/Harmfulness Before/After Downstream Fine-tuning.**
> ||||||$~~~~~$5K||||$~~~~~$1K||
> | -: | :-: | -: | -: | -: | -: | -: | - | -: | -: | -: |
> |||None|Full (7.7K)|High-Sim|Random|Low-Sim| |High-Sim|Random|Low-Sim|
> |**Initial**|Utility|6.93|6.68|7.01|7.28|7.11||6.78|7.03|6.91|
> ||GPT Score|3.62|1.01|1.04|1.05|1.04||1.30|1.28|1.43|
> ||GPT ASR|62.69%|0.19%|0.87%|0.96%|0.91%||6.83%|20.33%|10.19%|
> ||||||||||| |
> |**List**|GPT Score|4.55|4.80|4.78|4.77|**4.72**||4.87|4.93|**4.82**|
> ||GPT ASR|85.38%|92.50%|89.81%|90.38%|**88.85%**||94.42%|96.15%|**92.88%**|
> |**Pure Bad**|GPT Score|4.81|4.53|4.70|4.61|**4.50**||4.82|4.81|**4.78**|
> ||GPT ASR|92.31%|84.81%|89.04%|87.31%|**84.23%**||92.50%|92.31%|**91.15%**|
> ||||||||||| |
> |**Alpaca**|Utility|5.75|5.96|6.89|6.04|6.78||6.14|6.31|5.99|
> ||GPT Score|3.37|1.95|2.82|2.69|**2.48**||2.60|2.57|**2.28**|
> ||GPT ASR|57.31%|23.08%|45.19%|41.35%|**35.38%**||38.27%|36.54%|**30.58%**|
> |**SAMSum**|Utility|40.21%|51.02%|50.09%|51.16%|50.31%||45.49%|50.30%|51.22%|
> ||GPT Score|4.74|3.78|4.21|4.15|**4.10**||4.80|4.60|**4.44**|
> ||GPT ASR|92.50%|68.27%|81.73%|78.27%|**76.54%**||94.23%|89.23%|**84.42%**|
>
>
>
> **4. (Different Guardrail Forms)**  We are not entirely sure about the reviewer's reference to “*other forms of guardrails.*” In our experiments, we focused on preventing generating harmful content covered in BeaverTails [4] and evaluating the model’s safety using standard benchmarks (e.g., HEx-PHI, AdvBench). For instance, the evaluation questions from HEx-PHI already include 11 different types of harm, such as Illegal Activity, Fraud Deception, Tailored Financial Advice, etc. Although the effectiveness of the safety guardrail largely depends on the alignment dataset itself, our work focuses on building a more durable safety guardrail and understanding its mechanism. We would be happy to provide more experimental results in the revision if the reviewer could give more details.
>
> **5. (Data Contamination Detection)** This is an interesting idea! We conducted a preliminary experiment to test whether the membership inference method could reveal if such fine-tuning datasets have been used in upstream pretraining or alignment [1, 2]. We have surveyed some works and found that Shi et al. [1] proposed MIN-K% PROB, stating that “*an unseen example is likely to contain a few outlier words with low probabilities under the LLM.*” However, our results (available in Appendix C2) indicated that each dataset (including Pure Bad, Alpaca subsets, and Dolly subsets) has a low probability of being part of the Llama-2-7b-Chat training data. While this finding aligns with expectations (considering the dataset's release post-model release), as mentioned earlier, we view this as a potential threat to craft a harmful but benign-looking dataset.
>
> *References:*
>
> [3] Qi et al. Fine-tuning Aligned Language Models Compromises Safety, Even When Users Do Not Intend To! (ICLR 2024)
>
> [4] Ji et al. BeaverTails: Towards Improved Safety Alignment of LLM via a Human-Preference Dataset (NeurIPS 2023 Datasets and Benchmarks)

---

> > ### Comment · Reviewer_ykti · 2024-11-27
> > **Response to authors**
> >
> > I thank the authors for their rebuttal that addresses almost all my concerns.
> >
> > - About the types of guardrails, the authors seem to consider only one type of guardrail. There are many types:
> > [1] https://arxiv.org/abs/2407.05557
> > [2] https://github.com/NVIDIA/NeMo-Guardrails
> > [3] https://www.guardrailsai.com/
> >
> > In any case, I think the response is informative and accurate enough it is already worth increasing my score to 6.
> > Regards.

---

> > > ### Author Response · Authors · 2024-11-27
> > >
> > > We thank the reviewer for the encouraging response! We are glad our response addressed all your concerns. We have revised our paper to include other forms of guardrails as potential areas for exploration in future work. We appreciate your constructive feedback, which has helped us strengthen our work.
> > >
> > > Thank you once again for your thoughtful review and for recognizing the contributions of our study.

---

### Meta-Review · Area_Chair_U97f · 2024-12-20

**Metareview:**

This paper shows that keeping training data private and diversifying the training data improve the robusntess of safety guardrails for LLMs. It is also found that when the downstream tasks and the training data are more dissimilar, the guardrails are more robust.

To me, the primary weakness of the paper is overgeneralized claims without comprehensive evaluations. This point was also raised by several reviewers. The presented findings are restricted to a single model architecture, particular choice of jailbreak method, and guardrails.  One reviewer also suggested that the similar findings can be found in the literature so the authors are encouraged to look into that.

**Additional Comments On Reviewer Discussion:**

The primary concern on the lack of comprehensive evaluation remains after rebuttals.

---

### Decision · Program_Chairs · 2025-01-22

Reject